# Two Are Better than One: Context Window Extension with Multi-Grained Self-Injection

## Abstract

The limited context window of contemporary large language models (LLMs) remains a huge barrier to their broader application across various domains. While continual pre-training on long-context data is a straightforward and effective solution, it incurs substantial costs in terms of data acquisition and computational resources. To alleviate this issue, we propose SharedLLM, a novel approach grounded in the design philosophy of multi-grained context compression and query-aware information retrieval. SharedLLM is composed of two short-context LLMs such as LLaMA-2, termed upper model and lower model. The lower model functions as a compressor while the upper model acts as a decoder. The upper model receives compressed, multi-grained context information from the lower model and performs context-aware modeling on the running text. Information transfer between the compressor and decoder occurs only at the lowest layers to refrain from long forward paths in the lower model and redundant cross-attention modules in the upper model. Based on this architecture, we introduce a specialized tree-style data structure to efficiently encode, store and retrieve multi-grained contextual information for text chunks. This structure, combined with a search algorithm, enables rapid encoding and retrieval of relevant information from various levels of the tree based on the input query. This entire process, wherein the sender and receiver are derived from the same LLM layer, is referred to as *self-injection*. In our evaluation on long-context modeling and understanding tasks, SharedLLM achieves superior or comparable results to several strong baselines, striking an effective balance between efficiency and performance. Meanwhile, with the aforementioned design choices, SharedLLM can greatly reduce memory consumption, and demonstrates substantial speed-ups over other advanced baselines ($2\times$ over streaming, $3\times$ over encoder-decoder architectures). The core code of our implementation along with training and evaluation is available in appendix and supplementary.

## 1 Introduction

Since the release of GPT-3 (Brown, 2020), the rapid advancement of large language models (LLMs) (Chowdhery et al., 2022; Achiam et al., 2023; Touvron et al., 2023a;b; Dubey et al., 2024) has revolutionized the NLP research community and transformed various workflows. Pretrained on trillions of tokens, LLMs exhibit remarkable abilities, such as completing unfinished text or code and following human instructions to perform designated tasks after minimal supervised fine-tuning (Wei et al., 2021; Chung et al., 2024).

Despite their impressive capabilities, several factors limit their broader application. One major constraint is the *context window* size (Hsieh et al., 2024), which refers to the maximum number of tokens an LLM can process smoothly in a single input. The length of context window is typically set during pretraining—for example, LLaMA and LLaMA-2 have context windows of 2,048 and 4,096 tokens, respectively. When input text exceeds this limit, LLMs may exhibit erratic behavior during inference. Unfortunately, due to GPU memory constraints, high training costs, and the scarcity of long-context training data, LLMs are often pretrained with relatively short context windows. This limitation severely restricts their use in many daily tasks of large context lengths, such as long document summarization and information retrieval, where much longer windows are needed.

Many researchers endeavour to extending the context window of LLMs while minimizing the time, memory, and training costs during both training and inference. One approach involves post-pretraining LLMs on long-context corpora using hundreds of GPUs (TogetherAI, 2023; Xiong et al., 2024). Another line of work explores position interpolation (Chen et al., 2023; Peng et al., 2023), which rescales the RoPE (Rotary Position Embedding) frequency and attention scores. While this method is canonical, it still requires long-context continual pretraining. For example, YaRN (Peng et al., 2023) extends LLaMA's context length to 128K tokens by continuing pretraining on 64K-token sequences using full attention. The combination of parameter-efficient fine-tuning (PEFT) and sparse attention (Chen et al., 2024) accelerates tuning but faces challenges with extrapolation. Other approaches like streaming-style architectures (Xiao et al., 2024b; Zhang et al., 2024a; Yen et al., 2024), maintain a constant-sized memory that operates as a sliding window. While this design significantly reduces memory usage, its specialized attention pattern causes incompatibility with high-performance attention implementations like FlashAttention (Dao et al., 2022; Dao, 2023), potentially leading to slower inference speeds. Context compression techniques are also widely explored (Zhang et al., 2024a; Yen et al., 2024). Although they offer high parallelism, improving speed, they tend to consume significant memory, greatly limiting their real applications.

To strike a balance between efficiency and performance, we propose SharedLLM in this paper. SharedLLM features a lightweight hierarchical architecture that consists of one *upper model* and one *lower model*. Both models are initialized from the same *off-the-shelf* checkpoint of a short-context LLM, either in full or in part. Since there is no disparity in the hidden space between the two submodules, SharedLLM can be trained from scratch without extra stages for hidden-space alignment. The lower model compresses past context information into multi-grained representations. Through layer-wise connections between the lower and upper model, the compressed information can be passed to the upper model for context-aware language modeling.

For better organization and utilization of this multi-grained information, we further propose a dedicated data structure, dubbed *context tree*. The trees operate in parallel, with each tree handling an independent text chunk split from the longer raw input. The tree nodes contain token sequences of varying lengths, which are encoded and downsampled by the lower model to obtain a set of meaningful representations. The sequence corresponding to each node is split from its parent, making it a subsequence of all its ancestors. Meanwhile, we set the compression ratio proportional to the sequence length. As a result, the nodes at higher levels have longer sequence and larger compression ratio, which express more coarse-grained information after encoding, as opposed to nodes at lower levels. We further develop an algorithm for dynamic construction and search on the tree given the task-specific query. The algorithm accelerates the information gathering process by identifying the most relevant text segments and encoding them as fine-grained representations, while converting less relevant parts into coarse-grained representations in a depth-first manner. In the information injection stage, since the obtained context trees are ordered, we define a rule to assign chunk-level position ids for these encoded keys from the lower model, as well as the queries from the upper model. With these unique chunk-level ids, queries can perceive the relative position of trees to generate high-quality output.

With these design principles, SharedLLM presents extraordinary performance on down-stream tasks with high efficiency. Specifically, on language modeling tasks, trained on text of up to 8K tokens, our model demonstrates excellent extrapolation capabilities when tackling sequences up to 128K-token length. On other long-context instruction-following tasks with input lengths ranging from thousands to millions of tokens, SharedLLM delivers promising results comparable to several strong baselines. In terms of system indicators, all experiments with a maximum length of over 200K can be conducted on a single A800 80G GPU. SharedLLM delivers several times the speed of all baselines while maintaining relatively low memory consumption.

## 2 RELATED WORK

**Long-context Language Models.** There are two prevalent routines to build LLMs that are capable of processing extremely long text: directly pretraining on large corpus of targeted context length from scratch (Touvron et al., 2023a; Dubey et al., 2024; Jiang et al., 2023; GLM et al., 2024) or adapting short context-window LLMs to longer context lengths via combined various techniques (Tworkowski et al., 2024). The former approach consumes tremendous data and computational resources, while

the latter allows for more convenience and flexibility for researchers and developers to explore potential optimization to the default settings (Fu et al., 2024). The core idea behind these adaptations is to *mimic* short input scenarios (*i.e.*, length within the model's text window) when the input length exceeds window size. Attention map manipulation is the most common approach for this goal, which can be realized via positional encoding (PE) rescaling, such as ALiBi (Press et al., 2021), positional interpolation (PI) (Chen et al., 2023) and YaRN (Peng et al., 2023), or positional index rearranging (Xiao et al., 2024b; Ding et al., 2023; An et al., 2024; He et al., 2024). Both directly or indirectly adjust attention scores to be similar as the short-input scenarios so that the model can handily deal with. Another line of works compress past tokens sequentially into dense representations (Chevalier et al., 2023; Zhang et al., 2024a) as input at the next step or store them in an *external* retrievable memory (Wu et al., 2022; Xiao et al., 2024a) to reduce the input lengths. Yen et al. (2024) utilizes small model such as RoBERTa (Liu, 2019) for context encoding to boost speed and enable higher parallelism. However, this heterogeneous architecture necessitates meticulous task design for the extra pretraining and warmup stages to stabilize the fine-tuning process. In contrast to these works, our method directly tunes *off-the-shelf* models to compress context into structural representations for query-aware retrieval. Powered by efficient architecture design and a fast-forwarding mechanism, the whole procedure can be fully paralleled online without excessive memory usage, which greatly cuts down the latency during inference time.

**Efficient Methods for Long-context Modeling.** In vanilla self-attention, the space and time complexity grows quadratically ($O(L^2)$) with the input sequence length $L$, which can cause out-of-memory ($OOM$) issues on GPU clusters. A straightforward solution is to add parameter efficient fine-tuning (PEFT) modules (Chen et al., 2024; Zhang et al., 2024a;b) to shrink the size of gradient tensors during backward propagation. Many works strive to reduce the memory footprint of attention computation to enhance computational efficiency. Longformer (Beltagy et al., 2020) introduces a hybrid attention pattern to capture local and global semantic features concurrently. Katharopoulos et al. (2020) designs linearized attention that merely demands $O(L)$ space to accomplish attention computation. FlashAttention (Dao et al., 2022; Dao, 2023) and PagedAttention (Kwon et al., 2023) maximize the memory efficiency from system's perspective. More recently, Xiao et al. (2024b) discovers the "attention sink" phenomenon and proposes streaming-llm to address high perplexity issue in generation under window-attention. Our work basically follows the efficient design principle in three aspects: 1) lightweight architecture through lower layer self-injection; 2) compact structural representations via structural information extraction and compression; 3) efficient construction and retrieval algorithm based on the proposed context tree structure.

## 3 METHOD

In this section, we first introduce the overall architecture of our proposed SharedLLM in Sec. 3.1, and then elaborate on its two main components, lower model and upper model in Sec. 3.2 and 3.3.

### 3.1 OVERVIEW

As illustrated in Figure 1, SharedLLM adopts a hierarchical architecture, akin but not identical to classical encoder-decoder models. The *lower model*, or the "compressor", breaks down the long input context $X_C$ into smaller chunks that can be processed within limited GPU memory. It then uses the same LLM model to compress each context chunk into compact and structured representations in parallel. The *upper model*, or the "decoder", takes the rear part of the input text (the running context, such as questions) as input, then integrates the compressed information from the lower model, and finally predicts future tokens in an auto-regressive manner.

The lower and upper models are connected via shared key-value (KV) states and cross-attention modules between corresponding layers. To enable efficient and effective information retrieval and integration, the context information processed by the lower model is organized into a binary tree, referred to as the *context tree*, which stores multi-grained information at different levels. This structure allows the upper model to leverage its processed running text to efficiently retrieve relevant information from the binary tree based on a depth-first search algorithm. The retrieved information is then integrated with the input through cross-attention, enabling the model to answer questions or perform language modeling.

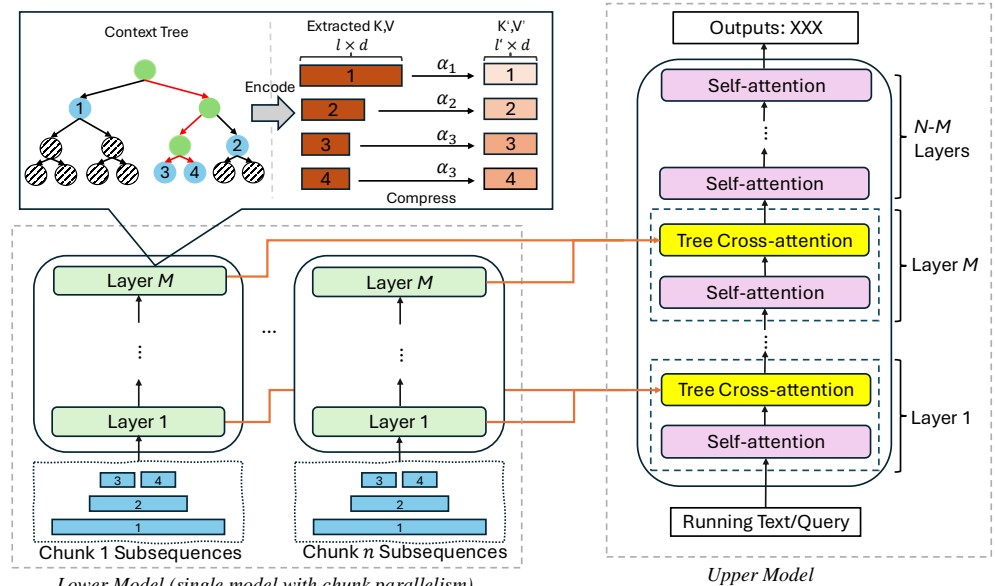

Figure 1: Overview of SharedLLM. It resembles general encoder-decoder architecture like T5 (Raffel et al., 2020). However, the interaction between lower and upper model occurs in the bottom $M$ layers through shared KV states which are encoded and compressed from the context tree nodes (subsequences). The orange arrows mark the paths through which KV states from lower model's self-attention layers are dispatched to the upper model's corresponding cross-attention layers, i.e., the process of *self-injection*.

In the following, we elaborate on the lower and upper model. To begin with, we first define some notations to enhance clarity and readability. Let $X = \{x_1, x_2, ..., x_T\}$ represent the entire input sequence, where $T$ denotes the sequence length. In comply with previous setting (Yen et al., 2024), we split these tokens into two continuous parts: $X = \text{concat}([X_C; X_D])$, where the past context $X_C$ and the running text $X_D$ serve as inputs to the lower and upper models, respectively. Moreover, the past context $X_C$ is further divided into $n$ smaller and non-overlapping chunks denoted by $C_1, C_2, ..., C_n$, namely, where $C_1 \cup C_2 \cup ... \cup C_n = X_C$ and $C_i \cap C_j = \emptyset, \forall i \neq j$. The chunk size is controlled to fit within the lower model's context window—e.g., 4,096 tokens for LLaMA-2-7B (Touvron et al., 2023b)—allowing the lower model to fully utilize its encoding capacity.

## 3.2 LOWER MODEL

The lower model is a small pretrained LLM, implemented as the first $M$ shallow layers of LLaMA-2. It independently encodes and compresses each past context chunk $C_i$ from the set of chunks $\{C_i\}_{i=1}^n$, and constructs a context tree that stores multi-grained information across various levels. The encoding for all chunks $\{C_i\}_{i=1}^n$ is fully paralleled to boost the speed. Below, we detail the context tree structure and its efficiency-enhanced query-dependent dynamic construction, and the tree search process.

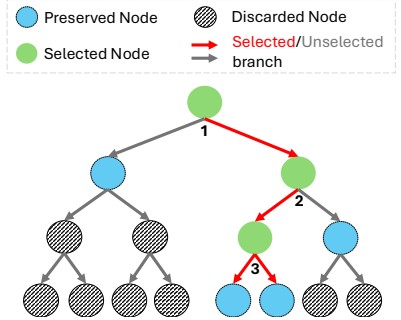

Figure 2: An running example of our tree (depth=3). The digits mark the step indices in the split-and-search procedure.

**Context Tree.** The motivation to build the context tree is intuitive and problem-driven. Given a text chunk $C_i$ and a task-specific query, the task-related information is often distributed unevenly across the chunk of text. For instance, to summarize a given passage, one should pay more attention to the topic sentences, collect messages from them and rephrase to produce the answer, rather than focuses much on narrative

details. Whereas in the task of passkey finding, detailed relations are more important than theme paragraphs. To this end, we aim for the contextual representations to capture fine-grained details for the relevant portions of the text, while encoding only coarse-grained information for the less relevant parts. The tree structure is the best fit to simulate this process: the spltting of nodes resembles splitting larger text chunks into smaller ones, from which we can get more fine-grained information.

In the context tree, its root node contains the entire chunk $C_i = \{x_s, ..., x_t\}$ where $x_p$ $(s \le p \le t)$ denotes a token, $s$ and $t$ are the start and end index of that chunk; and each other node consists of a sub-sequence of the chunk $C_i$. Then we introduce how to build the child nodes from a parent node. Specifically, for any non-leaf node that contains $l$ tokens $\{x_{u+1}, ..., x_{u+l}\}$, at training phase, we split it into two sub-sequences for constructing its left child and right child as:

$$C_{\texttt{parent}} = \{x_{u+k}\}_{k=1}^l, \quad C_{\texttt{left}} = \{x_{u+k}\}_{k=1}^b, \quad C_{\texttt{right}} = \{x_{u+k}\}_{k=b+1}^l. \tag{1}$$

Here we adopt a random splitting by setting $b = \lfloor \frac{l}{2} - \epsilon \rfloor$ and $\epsilon \sim \mathcal{N}(0, \sigma^2)$ where $\sigma$ is a predefined hyperparameter, since random lengths can slightly improve the performance as concluded in Zhang et al. (2024a). At test time, the noise $\epsilon$ is fixed to zero. One can continue this process until arriving at the limited tree depth. Next, building upon this static tree, we construct a more efficient query-dependent dynamic tree.

**Query-Dependent Dynamic Tree Construction and Search.** A task-specific query is typically highly relevant to certain tree nodes while being less relevant to others. For highly relevant nodes, further expansion is necessary to extract fine-grained information. However, for less relevant nodes, expansion is unnecessary. Thus, instead of constructing an entire static context tree as aforementioned, we build a query-dependent dynamic tree that expands only the relevant nodes, as shown in Figure 2, significantly saving both GPU memory and time.

Starting from the root node, we perform a depth-first splitting and search process. Each node sequence is first split into two subsequences according to Eq. (1). We then use a non-parametric policy $\pi$ to decide the next selected node based on the two subsequences, $\boldsymbol{x}_{\texttt{left}}$ and $\boldsymbol{x}_{\texttt{right}}$, and a query sequence $\boldsymbol{y}$:

$$\pi((\boldsymbol{x}_{\texttt{left}}, \boldsymbol{x}_{\texttt{right}}), \boldsymbol{y}) \rightarrow \texttt{left} \text{ or } \texttt{right}, \tag{2}$$

Here the policy $\pi$ determines whether the left or right child of the node will be selected. The unselected sibling node is marked as "preserved" and will not be expanded further. Note, the root node is always selected to ensure expansion. For policy $\pi$, it is task-specific. Specifically, regarding language modeling task, since there are no explicit queries (i.e., $\boldsymbol{y} = \emptyset$), we simply set $\pi$ to be deterministic:

$$\pi((\boldsymbol{x}_{\texttt{left}}, \boldsymbol{x}_{\texttt{right}}), \boldsymbol{y}) \equiv \texttt{right}. \tag{3}$$

For instruction-following tasks, such as question-answering, where queries like questions are available, $\pi$ selects the node with higher similarity to the query in the hidden space:

$$\pi((\boldsymbol{x}_{\texttt{left}}, \boldsymbol{x}_{\texttt{right}}), \boldsymbol{y}) = \underset{\phi \in \{\texttt{left}, \texttt{right}\}}{\arg\max} (\mathbf{sim}(\boldsymbol{h}_{\boldsymbol{x}_\phi}, \boldsymbol{h}_{\boldsymbol{y}})), \tag{4}$$

where $\mathbf{sim}(\cdot, \cdot)$ represents the cosine similarity between two vectors. The hidden vector $\boldsymbol{h}$ at the last position of a sequence is embedded by either the lower or upper model. Specifically, this involves a short forward pass through one self-attention layer in the lower model for $\boldsymbol{h}_{\boldsymbol{x}_\phi}$ and the upper model for $\boldsymbol{h}_{\boldsymbol{y}}$. Once the selected node is determined, the process continues with that node, repeating the procedure until reaching leaf nodes. At this point, both the left and right child are marked as "preserved".

For each preserved node, we feed its associated context into the lower model to obtain a collection of key-value (KV) states from all $M$ layers, denoted as $\mathbf{S} = \{\mathbf{K}, \mathbf{V}\}$, where $\mathbf{K}, \mathbf{V} \in \mathbb{R}^{M \times l \times d}$ represent the key and value states for all $M$ layers. Here, $l$ is the sequence length, and $d$ is the hidden dimension. Next, we perform a uniform downsampling along the length dimension to retain only a portion of the KV states, resulting in $\mathbf{S}' = \{\mathbf{K}', \mathbf{V}'\}$, where $\mathbf{K}', \mathbf{V}' \in \mathbb{R}^{M \times l' \times d}$ and $l'$ is the downsampled length. The compression ratio $\alpha$ for the node is defined as $\alpha = l/l'$. For the context tree, we apply a constant compression ratio $\alpha_w$ for all preserved nodes at level $w$, but the ratio diminishes progressively from top to bottom, i.e., $\alpha_w > \alpha_{w+1}$. In our implementation, we set $\alpha_w = 2\alpha_{w+1}$. Specific value of $\alpha_w$ can be found in Appendix A.1. This approach creates *coarse-to-fine* distribution

of semantic information from top to down: nodes at higher levels possess longer subsequences and are compressed with a higher compression ratio, corresponding to more coarse-grained information, while on the contrary, nodes closer to the bottom stores fine-grained information.

The overall compression ratio $\beta$ of a tree is defined as the ratio of the chunk length $|C|$ to the total length of the compressed KV states:

$$\beta = \frac{\sum l_w n_w}{\sum l'_w n_w} = \frac{|C|}{\sum l'_w n_w} \tag{5}$$

where $n_w$ is the number of preserved nodes at level $w$, and $l'_w$ is the compressed length of each preserved node at level $w$. For the convenience of parallel processing, we set $\beta$ same for all $n$ context trees. Experimental results in Section 4 demonstrate that this compression ratio can reach as high as 8, significantly improving efficiency.

### 3.3 UPPER MODEL

The upper model mainly inherits from the LLaMA architecture, which consists of $N$ (32 for LLaMA-2-7B) self-attention layers with slight modifications. As illustrated in Figure 1, for each one of the $M$ shallow layers, we add a cross-attention module on the top of the vanilla self-attention layer for information fusion.

**Position-aware Cross-attention on the Context Tree.** In Section 3.2, we can obtain a sequence of tree-structural representations $\mathcal{S}' = \{\mathbf{S}'_1, ..., \mathbf{S}'_n\}$ for $n$ chunks $\{C_i\}_{i=1}^n$, where $\mathbf{S}'_i = \{\mathbf{K}'_i, \mathbf{V}'_i\}$ stands for the representations of chunk $C_i$. Since the sequence of chunk keys $\mathcal{K} = \{\mathbf{K}'_1, ..., \mathbf{K}'_n\}$ is produced from ordered chunks $\{C_1, ..., C_n\}$, their positional information should be aware at chunk level by the query. We assign the following chunk-level positional indices to $\mathbf{Q}$ and $\mathcal{K}$:

$$\mathbf{P_Q} = \{\underbrace{n, n, ..., n}_{|X_D|}\}, \quad \mathbf{P}_\mathcal{K} = \{\underbrace{0, 0, ..., 0}_{|C_1|/\beta}, \underbrace{1, 1, ..., 1}_{|C_2|/\beta}, \underbrace{n-1, n-1, ..., n-1}_{|C_n|/\beta}\}. \tag{6}$$

Here we view the upper model's query $\mathbf{Q}$ as one chunk and endow with the largest positional index, because $\mathbf{Q}$ is encoded from $X_D$ which is behind all context chunks $X_C$ in the raw input sequence $X$. We will show in Section 4.4 that this setting also facilitates chunk-level extrapolation and answer text production in downstream tasks.

We then conduct cross attention between the query $\mathbf{Q}$ and concatenated KVs to integrate their carried context information into running context for more coherent language modeling:

$$O = \text{cross\_attn}(\mathbf{Q}, \text{concat}([\mathbf{K}'_1; ...; \mathbf{K}'_n]), \text{concat}([\mathbf{V}'_1; ...; \mathbf{V}'_n])). \tag{7}$$

**Training** We use the standard language modeling loss during training, which maximizes the log probability of the ground-truth tokens in the target sequences $X_{\text{tar}}$, conditioned on the context $X_C$ and all preceding tokens $x_{<t}$ from $X_D$:

$$\mathcal{L} = -\sum_{x_t \in X_{\text{tar}}} \log P(x_t | X_C; x_{<t}). \tag{8}$$

For language modeling data, $X_{\text{tar}} = X_D$, i.e., the target tokens are all tokens in $X_D$, excluding the first token. For instruction-following data, $X_D$ includes both the instruction $X_{\text{inst}}$ and the annotated response $X_{\text{res}}$. In this case, we set $X_{\text{tar}} = X_{\text{res}}$, meaning that we optimize only for the response tokens, while the instruction text is masked during loss calculation.

## 4 EXPERIMENTS

### 4.1 SETUP

**Initialization** We initialize the upper model with LLaMA-2-7B in language modeling and LLaMA-2-Chat-7B in supervised fine-tuning (SFT), in consistent with previous works (Chen et al., 2024;

Table 1: Perplexity of models trained on mixed dataset. "OOM" means out-of-memory exception raised during inference. Excessively large perplexities ($> 10^2$) are hidden with a dash ("-").

| Model | PG19 | | | | ProofPile | | | | CodeParrot | | | |
|---|---|---|---|---|---|---|---|---|---|---|---|---|
| | 4K | 16K | 32K | 100K | 4K | 16K | 32K | 100K | 4K | 16K | 32K | 100K |
| StreamingLLM | 9.21 | 9.25 | 9.24 | 9.32 | 3.47 | 3.51 | 3.50 | 3.55 | 2.55 | 2.60 | 2.54 | 2.56 |
| AutoCompressor | 11.80 | - | - | OOM | 4.55 | - | - | OOM | 3.47 | - | - | OOM |
| LongAlpaca-16K | 9.96 | 9.83 | - | OOM | 3.82 | 3.37 | - | OOM | 2.81 | 2.54 | - | OOM |
| LongLlama | 9.06 | 8.83 | OOM | OOM | 2.61 | 2.41 | OOM | OOM | 1.95 | 1.90 | OOM | OOM |
| LongChat-32K | 9.47 | 8.85 | 8.81 | OOM | 3.07 | 2.70 | 2.65 | OOM | 2.36 | 2.16 | 2.13 | OOM |
| Activation Beacon | 9.21 | 8.34 | 8.27 | 8.50 | 3.47 | 3.34 | 3.32 | 3.31 | 2.55 | 2.43 | 2.41 | 2.62 |
| SharedLLM | **8.98** | **8.15** | **7.96** | **8.24** | **3.36** | **3.24** | **3.21** | **3.19** | **2.33** | **2.25** | **2.25** | **2.36** |

Table 2: Perplexity of models trained on downsampled RedPajama. LLaMA-2-32K and YaRN-2-128K have seen sequence as long as up to 32K and 64K tokens respectively at training time, while CEPE and SharedLLM are trained on 8K-token sequences. †: results run on the reproduced model following original paper and the released code. Notations share the same meanings with the last table.

| Model | Arxiv | | | | PG19 | | | | ProofPile | | | |
|---|---|---|---|---|---|---|---|---|---|---|---|---|
| | 4K | 8K | 32K | 128K | 4K | 8K | 32K | 128K | 4K | 8K | 32K | 128K |
| LLaMA-2-7B (4K) | 2.60 | - | - | OOM | 6.49 | - | - | OOM | 2.28 | - | - | OOM |
| *Books3 involved in training* | | | | | | | | | | | | |
| LLaMA-2-32K | 2.60 | 2.51 | 2.32 | OOM | 6.61 | 6.50 | 6.97 | OOM | 2.46 | 2.22 | 2.27 | OOM |
| YaRN-2-128K | 3.13 | 2.96 | 2.34 | OOM | 6.15 | 6.02 | 6.32 | OOM | 2.70 | 2.47 | 2.41 | OOM |
| CEPE | 2.86 | 2.84 | 2.34 | 2.91 | 6.60 | 6.24 | 6.66 | 5.99 | 2.22 | 2.33 | 2.26 | 2.23 |
| *Books3 not involved in training* | | | | | | | | | | | | |
| CEPE† | 3.03 | 3.02 | 2.51 | 2.97 | 6.69 | 6.40 | 6.80 | 6.10 | 2.38 | 2.43 | 2.45 | **2.39** |
| SharedLLM | **2.99** | **2.97** | **2.46** | **2.91** | **6.59** | **6.31** | **6.72** | **6.00** | **2.36** | **2.37** | **2.41** | 2.46 |

Yen et al., 2024; Zhang et al., 2024a). The lower model is initialized with the weights of bottom $M$ layers from the same checkpoint as the upper model, where we set $M = 4$ in language modeling and $M = 16$ in SFT.

**Dataset** For language modeling, we follow Yen et al. (2024) to prepare the training data by sampling a subset of 20B (1%) tokens from RedPajama's all 7 domains (Together, 2023). Due to the copyright issue, the books3 subset in Books domain (books3 + PG19) is unavailable and thus excluded from our training set, yet we do not renormalize sampling probability across domains. As a result, the proportion of PG19 increases in the final dataset compared with the default setting in Touvron et al. (2023b). The sampled texts are truncated to 8,192 tokens for training. In SFT, we follow Zhang et al. (2024a) to use the same mixed dataset composed of downsampled RedPajama and LongAlpaca (Chen et al., 2024), where the input length is filtered to range between 1200 to 8192 tokens, following Zhang et al. (2024a).

**Training** We train SharedLLM on an $8\times$ A800 GPU machine. The batch size is set to 1 per GPU with gradient accumulation of 16 steps (global batch size is 128) for language modeling and 1 step (global batch size is 8) for SFT. Zero Redundancy Optimizer (ZeRO) stage 3 from DeepSpeed without offload is enabled in both training to distribute the memory allocation among GPUs. The cross-attention layers remain fully tunable, while we opt to train upper model's top $N - M$ self-attention layers in language modeling as post-injection aggregation for faster convergence. No parameter efficient fine-tuning (PEFT) techniques, such as LoRA, are applied during both training, as PEFT seriously slows down model's convergence (Chen et al., 2024), which actually requires longer tuning time than partial parameter fine-tuning to reach the optimum. We adopt AdamW optimizer with the starting learning rate $1e^{-5}$ and cosine scheduler during training. The chunk size is set to 1,024 for langauge modeling or 512 in SFT, with tree height $h = 3$ and compression ratio $\beta = 8$. For other configurations and hyperparameters, please refer to Appendix A.1 for more details.

## 4.2 MAIN RESULTS

**Language Modeling.** We first evaluate our models on the language modeling task with sequence lengths ranging from 4K to 128K using a single A800 80GB GPU. The evaluation covers four datasets: ArXiv, PG19 (Rae et al., 2020), ProofPile (Azerbayev et al., 2024), and CodeParrot Tunstall et al. (2022) under two settings that utilize different training datasets. Under each setting we test on three out of the four datasets, respectively. The results are posted on Table 1 and 2. All perplexity values in these tables are averaged over 100 examples except for the 128K length, on which we test only 10 examples due to the data scarcity (Yen et al., 2024; Zhang et al., 2024a). For the experiments on encoder-decoder and hierarchical models at 4K length, the input is divided by half (2K/2K) and fed separately into their two submodules. The results show that our model owns strong extrapolation capability—it avoids perplexity explosion even tested on 128K-token length although it only has seen up to 8K-token sequences during training. In Table 1, SharedLLM outperforms other baselines trained on mixed dataset 3-10%. In Table 2, for models trained on RedPajama, the perplexity without books3 (bottom) gets a bit worse than those including books3 in the training set, showing the major contribution to language modeling by books3, consistency with the discovery claimed in Yen et al. (2024). Notably, SharedLLM outperforms CEPE in nearly all cases except 128K context length on ProofPile, showcasing the effectiveness of structural self-injection mechanism. Between the two settings, the improvement over Activation-Beacon is more pronounced than over CEPE, because CEPE experiences an additional pretraining stage to adapt the RoBERTa encoder to the RedPajama corpus and a warmup stage to align the hidden space between encoder and decoder. In contrast, SharedLLM can directly be finetuned from publicly available *off-the-shelf* checkpoints, which saves a great amount of training efforts.

**Long-context Understanding Benchmarks.** We continue to test the supervised fine-tuned version of SharedLLM on many downstream tasks from InfiniBench (Zhang et al., 2024c) and Long-Bench (Bai et al., 2023). Both benchmarks consist of a variety of long-context tasks established from raw and synthetic datasets.

On InfiniBench, we are interested in the following two tasks: **Math.Find** asks a model to retrieve a special value specified in the prompt (e.g., minimum, maximum, medium, etc.), which examines both the precise retrieval and query understanding abilities of the model. **En.MC** instructs a model to collect key information from a extremely long passage and choose the correct answer from many candidate options. We compare SharedLLM with advanced baselines capable of extremely long inputs, as shown in Table 3. SharedLLM surpasses these strong

Table 3: Evaluation of different methods on **Math.Find** and **En.MC** from InfiniBench.

| Method | Math.Find | En.MC |
|---|---|---|
| LM-Infinite | 5.71 | 30.57 |
| StreamingLLM | 6.00 | 32.31 |
| InfLLM | 11.14 | 31.44 |
| SharedLLM | **13.58** | **33.65** |

baselines on both tasks (2.44 points or 21.9% on Math.Find, 1.34 points or 4.1% on En.MC over state-of-the-arts), showing excellent capabilities in tackling extremely long input.

For LongBench, We report the categorical scores on all 14 English tasks in 5 categories, including single-document QA (SD-QA), multi-document QA (MD-QA), summarization (Summ.), few-shot learning (FS) and code-completion (Code), as shown in Table 4. SharedLLM outperforms or matches other advanced instruction-tuned long-context baselines across all five categories. Particularly, we notice that MD-QA. We note that truncation from the middle could reduce the difficulty of some tasks and improve the performance (Zhang et al., 2024a), especially on decoder-only models, as relevant information for many tasks is located at the beginning or end of the entire text rather than the middle part.

## 4.3 TIME AND MEMORY EFFICIENCY

Apart from strong performance on downstream tasks, SharedLLM demonstrates high computational efficiency in terms of both speed and GPU memory utilization. We compare these metrics produced by SharedLLM against other representative models from the model classes of streaming (Zhang et al., 2024a), encoder-decoder (Yen et al., 2024) and vanilla Peng et al. (2023) architectures that have shown competitive performance in prior evaluations. The results are visualized in Figure 3.

Table 4: Evaluation of different methods on LongBench. Text samples are truncated to the test length from middle before generation in some models. We particularly highlight these values in "Test Length" column, as well as model's training length ("Train Length"). Models in the upper rows follow the conventional "pretrain+finetuned" paradigm while models in the bottom rows are directly trained on mixed dataset without continual pretraining to extend the context window in advance.

| Method | Train Length | Test Length | SD-QA | MD-QA | Summ. | FS | Code |
|---|---|---|---|---|---|---|---|
| Llama-2-7B-Chat | 4K | 4K | 24.90 | 22.60 | 24.70 | 60.00 | 48.10 |
| StreamingLLM | 4K | 4K | 21.47 | 22.22 | 22.20 | 50.05 | 48.00 |
| LongAlpaca-16K | 16K | 16K | **28.70** | 28.10 | **27.80** | **63.70** | 56.00 |
| YaRN-128K | 64K | 32K | 24.03 | 24.11 | 19.82 | 60.00 | **62.73** |
| Activation Beacon | 8K | 16K | 28.27 | 28.44 | 25.15 | 61.00 | 57.75 |
| SharedLLM | 8K | 32K | 28.15 | **30.93** | 24.28 | 63.50 | 57.95 |

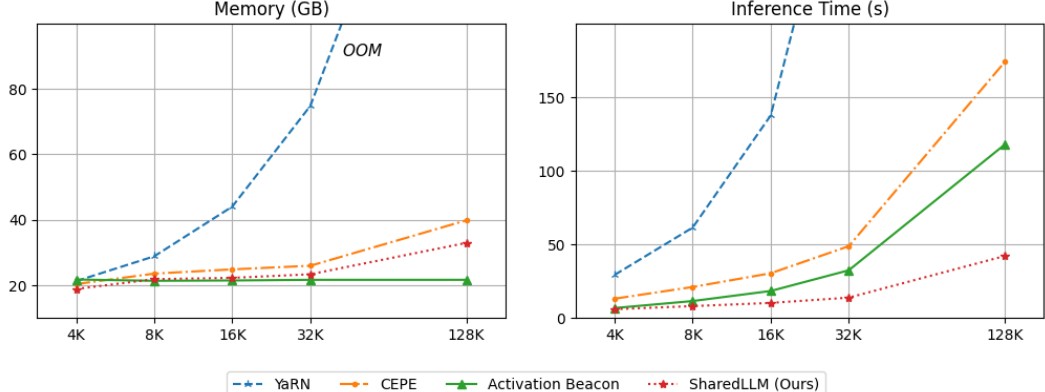

Figure 3: Comparison of memory usage (left) and total inference time on 100 examples (right) between SharedLLM and other recent baselines. The data is collected by running a tiny experiment on 100 examples in corresponding lengths. "OOM" means out-of-memory exception triggered during test time.

YaRN (Peng et al., 2023), which exploits the same fully attention as vanilla auto-regressive LLaMA, has $O(L^2)$ time and space complexity. The squared complexity makes it the only model that triggers out-of-memory exception at 128K length. Activation Beacon (Zhang et al., 2024a), which adopts the streaming processing paradigm, maintains a minimum constant memory $O(l)$ under different input lengths $L$, where $l$ is the sliding window length. However, Activation Beacon is incompatible with FlashAttention (Dao, 2023) also due to its specialized attention paradigm, which causes a sharp increment in inference time as input size grows. CEPE can process past context chunks in parallel, but these chunks must be passed through all its encoder layers (24-layer RoBERTa in CEPE) and layer-wise linear projections to obtain the final hidden states for cross-attention, leading to even slower inference speed than non-parallel Activation Beacon. In contrast, SharedLLM avoids such redundancy through shallow-layer compression and injection, which exhibits significant speed-up and limited memory consumption.

## 4.4 ABLATION STUDIES

We consider the following ablative settings to verify the rationale of the design considerations in SharedLLM: 1) tree depth; 2) compression ratio; 3) the collection of context information injection layers; 4) other configurations, including the effect from retrieval policy $\pi$ (only for instruction-following task), the noise in node splitting, and the addition of chunk-level positional indices during cross-attention.

The results are displayed in Table 5, from which we find that both tree depth and compression ratio should be set appropriately to achieve near-optimal performance. For example, SharedLLM performs best when the tree height is 3. If the height is too small, i.e., the tree is *undersplit* and the chunk

Table 5: Ablative Studies on different configurations of structural information injection. The best values in each category and settings consistent with our defaults are highlighted in **bold**.

| Item | Configuration | Arxiv (32K) | MD-QA |
|---|---|---|---|
| Tree Height | 2 | 2.51 | 30.15 |
| | **3** | **2.46** | **30.93** |
| | 4 | 2.57 | 29.47 |
| Compression Ratio $\beta$ | 1 | **2.43** | 30.55 |
| | 4 | 2.48 | 30.28 |
| | **8** | 2.46 | **30.93** |
| | 16 | 2.52 | 29.81 |
| Injection Layers | **Continuous Bottom** | **2.46** | **30.93** |
| | Continuous Top | 2.61 | 28.66 |
| | Interleaving | 2.57 | 29.15 |
| Other Settings | **Default** | **2.46** | **30.93** |
| | w/o retrieval | - | 29.27 |
| | w/o noise | 2.51 | 30.08 |
| | w/o chunk-level pid | 2.49 | 29.81 |

size is excessively large so that only coarse-grained context information is retained while task-related fine-grained information is not explicit, or too large, i.e., the tree is *oversplit* and the leaves carry fragmented information which can hardly provide valuable clues for task solving, performance degrades accordingly. A similar trend can be viewed on global compression ratio $\beta$. While abandoning downsampling KV ($\beta = 1$) may bring decline in perplexity, its query-aware information retrieval ability deteriorates. In terms of injection layer selection, our implementation, which is refer to as *continuous bottom*, injects the context information in the bottom $M$ layers. In contrary, *Continuous top* injects context information at the topmost $M$ layers (from layer $N - M + 1$ to layer $N$). Interleaving applies cross-attention at regular intervals, such as layer 4, 8, 12, 16... Among these configurations, SharedLLM wins over *continuous top* and *interleaving* on both tasks, indicating the correctness of injection layer selection in SharedLLM.

For other settings, as shown in the bottom rows, removing either of them causes performance drop compared to the default setting, which reveals the contributions of the three design considerations to model's performance. Among these items, the query-aware information retrieval is the core component for the context-tree so that the performance on MD-QA drops mostly after removing it from the network. The sequential order is similarly important and should be perceived during cross-attention to organize the answer accordingly.

Besides the effect on task performance, we also perform more experiments to explore how these configurations impact speed and memory in Appendix C.

## 5 CONCLUSION

In this work, we present SharedLLM, which leverages a self-injection mechanism to adapt a pair of short-context LLMs for efficient long-context modeling. By integrating the operations of context compression and key information retrieval into a dedicated binary-tree structure, SharedLLM excels in language modeling and various downstream instruction-following tasks, while maintaining excellent memory and time efficiency. Besides, SharedLLM is directly trained from off-the-shelf LLMs, eliminating the need for additional feature alignment steps and making implementation easier. We hope this learning paradigm can be generalized to other short-context LLMs, offering a scalable approach for a context-window extension to an arbitrary length.

**Limitations.** While SharedLLM demonstrates superior performance on both language modeling and long-context benchmarks, as well as high efficiency in terms of time and memory, there are still some limitations. First, although this work strikes a relatively good balance between efficiency and performance at the model architecture level, further improvements could be achieved by optimizing at the system and hardware levels. Second, while a simple and effective retrieval mechanism is implemented in this work, more advanced retrieval techniques, such as BM25 (Robertson et al.,

2009) and Graph-RAG (Edge et al., 2024), were not explored and may further enhance performance. We aim to pursue these improvements in future research.

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

# A  IMPLEMENTATION DETAILS

## A.1  TRAINING CONFIGURATIONS

We list more training configurations that are not specified in the main text in Table 6. The sequential values of $\alpha$ are level-wise compression ratio, from level 1 to level 3.

Table 6: Configurations for training on both tasks.

| Item | Language Modeling | Supervised Fine-tuning |
|------|:---:|:---:|
| training epoch | 1 | 2 |
| warmup ratio | 0.01 | 0.001 |
| $\sigma$ | l/5 | l/10 |
| chunk size | 1024 | 512 |
| $\alpha$ | $1/16, 1/8, 1/4$ | |
| AdamW ($\beta_1, \beta_2$) | 0.9, 0.999 | |

## A.2  ONLINE SPLIT-AND-SEARCH ALGORITHM

We provide the pseudo code for the online split-and-search algorithm introduced in Section 3.2, from the splitting of root node till collecting all key-value states for all preserved nodes and all $M$ layers. The code snippet in the entire model.py file can be found in the supplementary material.

**Algorithm 1:** Pseudocode of dynamic Construction-and-Search.

```
# N: number of trees; L: chunk size
# depth: tree depth; chunk_ids: the entire input ids for chunk in shape (N, L)
# gamma: a hyper-parameter to adjust the variance of the gaussian sampling

selected_input_ids = chunk_ids
selected_length = chunk_ids.shape[-1]
all_kvs = []

for i in range(depth):
    # sample lengths of left and right child
    if i < depth - 1:
        half_length = last_length // 2
        sigma = half_length / gamma
        delta = random.randn(1) * sigma
        l_left, l_right = half_length - int(delta), half_length + int(delta)

        # split the node into two children
        left_input_ids, right_input_ids = input_ids[:l_left], input_ids[-l_right:]
        # query_aware is a flag indicating if the selected nodes are determined on query
        if query_aware:
            # short forward (1-layer) to get representation vectors for the query and two nodes
            h_q = upper_model(query, 1)
            h_left, h_right = lower_model(left_input_ids, 1), lower_model(right_input_ids, 1)
            selected = argmax(sim(h_q, h_left), sim(h_q, h_right))
        else:
            selected = 1 # deterministic example, can change to 0 or random selection

        selected_input_ids = [left_input_ids, right_input_ids][selected]
        selected_length = [l_left, l_right][selected]

        preserved_input_ids = [left_input_ids, right_input_ids][1 - selected]
    else:
        preserved_input_ids = cat(last_input_ids.chunk(2, -1), 0)

    cur_level_kvs = lower_model(preserved_input_ids).past_key_values
    cur_level_kvs = downsample(cur_level_kvs)
    all_kvs.append(cur_level_kvs)
```

cat: concatenation; chunk: split into the specified number of chunks

### A.3 DATASET STATISTICS

**Downsampled Redpajama.** We follow Yen et al. (2024) and Touvron et al. (2023b) to prepare our training set. The proportions of data regarding seven domains in the resulted training set are listed in Table 7.

Table 7: Dataset composition in our downsampled Redpajama (20B) tokens.

| Domain | Proportion (%) |
|---|---|
| Arxiv | 2.5 |
| Books (w/o S3) | 4.5 |
| C4 | 15.0 |
| CommonCrawl | 67.0 |
| Github | 4.5 |
| StackExchange | 2.0 |
| Wikipedia | 4.5 |

**Mixed Dataset in SFT.** This dataset is directly picked from Zhang et al. (2024a), which is a mixture of RedPajama and LongAlpaca (Chen et al., 2024). We follow Zhang et al. (2024a) to only filter samples whose lengths range from 1K to 8K. The distribution of samples in terms of lengths is below.

Table 8: Proportion of samples within each length interval.

| Length | <2K | 2∼4K | 4∼6K | 6∼8K |
|---|---|---|---|---|
| Proportion | 47% | 29% | 8% | 16% |

## B MORE EXPERIMENTS

### B.1 RESULTS ON OTHER BASE LLMS

For LongBench, We report the results of SharedLLM that uses LLaMA-3-8B to initialize both lower and upper models, as shown in Table 9. Similar as the outcome using LLaMA-2-Chat-7B in Table 4, SharedLLM enhances all categorical performance on LongBench.

| Model | SD-QA | MD-QA | Summ. | FS | Code |
|---|---|---|---|---|---|
| LLaMA-3-8B | 15.12 | 7.95 | 26.13 | 68.75 | 56.04 |
| SharedLLM | 22.31 | 13.58 | 27.05 | 70.52 | 62.60 |

Table 9: Results on LongBench using LLaMA-3-8B as backbone LLM.

### B.2 PASSKEY RETRIEVAL

We further assess the retrieval capability of SharedLLM on passkey retrieval task, as known as needle-in-haystack (NIAH). Following the settings in Yen et al. (2024), we train a new version of SharedLLM that can perform accurate passkey retrieval from the haystacks of surrounded nonsense. We follow the examples in Chen et al. (2024) to set up the single key-value pair test cases. The results averaged on 10 random generated NIAH test samples are shown in Table 10. Both CEPE and SharedLLM distilled on 4K length can retrieve needles from much longer haystacks.

| Method | 4K | 8K | 16K | 32K |
|---|---|---|---|---|
| CEPE | 100 | 100 | 90 | 40 |
| SharedLLM | 100 | 100 | 100 | 60 |

Table 10: Needle-in-hay-stack on distilled version of SharedLLM.

## C  OVERHEAD ANALYSIS

In section 4.3, we have explained the outstanding efficiency of our model by comparing the memory usage and inference speed with other competitors. In this section, we give a more comprehensive analysis towards the inherent factors that may impact model's efficiency, including compression ratio $\beta$, tree height $h$, the number of shared layers $M$ and the retrieval-based policy which requires an additional short forward pass.

Table 11: Inference time under various $M$ with constant $h = 3$ and $\beta = 8$. Our default setting is highlighted in **bold**.

| $M$ | 1 | 2 | **4** | 8 | 16 |
|---|---|---|---|---|---|
| Time (s) | 6.78 | 9.35 | **11.81** | 16.81 | 25.85 |
| Memory (GB) | 21.04 | 21.50 | **22.39** | 24.08 | 27.82 |

We rerun our experiments to measure the forward time and memory cost from language modeling on 8K tokens, adjusting one variable at a time while keeping others at their default values. The results are shown in Table 11, 12 and 13. Among these factors, the number of injection layers, $M$, has the most significant impact on both speed and memory: both memory and latency grows as $M$ increases. As an opposite, compression ratio $\beta$ and tree height $h$ produces nuances effect on both metrics. For example, if we decreases $\beta$ from 64 to 1 (preserve all KVs), the inference time increases by 6.7% while memory increases by 3%. A similar trend is observed on experiments with tree height $h$. We speculate that the reason behind these outcomes are partly from the internal optimization in FlashAttention, which efficiently computes attention blockwisely. When the configuration meets its requirement for block size and hidden dimension (e.g., length is divisible by 256),

Table 12: Inference time under various $\beta$ with constant $h = 3$ and $M = 4$. Our default setting is highlighted in **bold**. For $\beta \in \{1, 2\}$, we are not able to set levelwise compression ratios and thus we set the compression ratio same as the $\beta$ for every level of the tree.

| $\beta$ | 64 | 32 | 16 | **8** | 4 | 2 | 1 |
|---|---|---|---|---|---|---|---|
| Time (s) | 11.68 | 11.73 | 11.78 | **11.81** | 11.87 | 12.04 | 12.47 |
| Memory (GB) | 22.20 | 22.20 | 22.20 | **22.39** | 22.40 | 22.35 | 22.97 |

We further investigate the potential overhead caused by the extra short forward path query-aware splitting-and-search algorithm. As shown in Table 14, we observe it incurs around 15% overhead in both time and space. We believe this type of overhead can be further eliminated with more careful optimization to the implementation details.

Table 13: Inference time under various $h$ with constant $\beta = 8$ and $M = 4$. Our default setting is highlighted in **bold**.

| $h$ | 1 | 2 | **3** | 4 |
|---|---|---|---|---|
| Time (s) | 11.16 | 11.55 | **11.81** | 11.86 |
| Memory (GB) | 19.72 | 22.42 | **22.39** | 22.41 |

Table 14: Comparison of time and memory consumption when query-based retrieval is incorporated/not incorporated in SharedLLM. $h$, $M$ and $\beta$ are fixed at the default values.

| Setting | Time | Memory |
|---|---|---|
| w/o query-aware retrieval | 11.81 | 22.39 |
| w query-aware retrieval | 13.18 | 25.44 |

