# OpenReview forum: "Two Are Better than One: Context Window Extension with Multi-Grained Self-Injection"
_ICLR.cc/2025/Conference — Submitted to ICLR 2025_

### Official Review · Reviewer_PZ2g · 2024-10-20

**Soundness:** 3
**Presentation:** 2
**Contribution:** 3
**Rating:** 5
**Confidence:** 3

**Summary:**

The paper introduces SharedLLM, an innovative approach designed to extend the context window of LLMs without incurring the substantial costs associated with continual pre-training on long-context data. The authors claim that SharedLLM achieves comparable or superior results on long-context tasks while significantly reducing memory consumption and increasing processing speed compared to existing methods.

**Strengths:**

1. The SharedLLM introduces an original dual-model architecture with a "context tree" data structure, enhancing the efficiency of context compression and retrieval for large language models.
2. Despite the complexity of the concepts introduced, the paper communicates the workings of the SharedLLM and its underlying mechanisms

**Weaknesses:**

1. The evaluation only uses the LLaMA-2 model, with no justification for not including more recent or varied models like LLaMA-3.
2. The method proposed in this paper does not seem to outperform other models in Longbench.

**Questions:**

See Weakness section

---

### Official Review · Reviewer_swMr · 2024-10-27

**Soundness:** 2
**Presentation:** 3
**Contribution:** 2
**Rating:** 3
**Confidence:** 5

**Summary:**

This paper proposes ShardLLM to reduce heavy training and inference cost in long-context LLMs. Specifically, this method insight is on the multi-grained context compression and query-aware information retrieval. SharedLLM uses  two short-context LLMs, where  the lower model functions as a compressor while the upper model acts as a decoder. The lower model divides the sequence into non-overlapping chunks and makes split-and-search procedure to choose relevant chunks. The selected chunks are then downsampled to reduce KV cache cost. The upper model encodes these keys and values via cross attention with chunk level positional embedding.

The paper makes experiments on language modeling and supervised fine-tuning to verify the effectiveness of SharedLLM. In language modeling, SharedLLM outperforms other efficient long-context baselines. In supervised fine-tuning, SharedLLM also achieves strong performance in LongBench and InfiniteBench. The efficient experiments show that SharedLLM is cheap and easy to deploy. Finally, the ablation studies discuss the design choice.

**Strengths:**

1. The paper focuses on an important research area, where LLMs are struggling to achieve effective and efficient long-context training and inference. SharedLLM integrates the compression and retrieval idea, which is a promising research direction in this field.

2. The paper is well written. Although the proposed method includes a long procedure, the method section is organized well to help readers get the core idea. The experiment section includes all the necessary part, including performance, efficiency, and ablation studies.

**Weaknesses:**

1. The method is very complicated. I admit that "complicated" itself is not sometimes a weakness. However, simple and elegant design is usually working well in model architecture research. Complicated design brings engineering difficulty and optimization problem. SharedLLM uses the last output as the embedding of a chunk which is used for retrieval by the queries. It is highly questionable whether optimization is valid where sparse computation usually has difficulty on gradient estimation.

2. SharedLLM divides context into different chunks and adds downsampling module to compress KV cache. This may hurt performance on some long-context tasks. The experiment part is not strong enough to support SharedLLM's long-context capability. I will give some existed benchmarks and easy examples accordingly:
- Needle-in-a-haystack is now a compulsory evaluation to show long-context's retrieval ability, which is also included in InfiniteBench. However, the experiment only includes two subtasks. The other part is also essential to show model's long-context ability.
- If I query the model to repeat all the previous context, the sparse-divided and compress context may hurt the context information. Moreover, If my first query does not include some context information, and my second query needs that again. Then, SharedLLM will drop it in the first response and can not answer my second question.

3. The experiment also lacks important baselines of KV pruning methods. For example, StreamingLLM is an early baseline which directly drops all the global information while only maintaining the local KV cache. There are many sparse KV cache works, including H2O, SnapKV, FastGen. These works are efficient and memory-friendly, while maintaining parts of long-context capability.

**Questions:**

1. I'm curious about the result in Table 1 and Table 2. CEPE in line 349 is a re-produced result. Does it mean that the other results are all from the public checkpoint of the according paper? If so, I think the perplexity comparison is meaningless due to different experiment setting. If not so, the perplexity in short context (4k) is highly different, which is also weird.

2. On Infinibench evaluation, why you are only interested in two subtasks? There is not a rationale for that. After all, if you use a bench for evaluation, you usually use the whole subtasks.

---

> ### Comment · Reviewer_swMr · 2024-11-24
>
> Thanks for your detailed response! I will continue discussing some details as follows:
>
> 1. "Complicated" is more like a subject judgement. You believe it is simple and straight only because that's your invention. Admittedly, it is not compulsory in an academic paper. You can consider it as my personal feeling and advice, which won't affect the rating.
>
> 2. I still do not agree about your comparison view in the experiment part. Even though some KV pruning methods are post-training technique, long-context continue-training is not a difficult part in 2024 year [1, 2]. If I consider your method as an aligned training+inference method, we can also split the two parts.
>
> 3. Following 2, only fairly comparing your method with CEPE is not solid enough academically. The baseline comparison needs to be done by yourself, since there are almost contemporary work. I think listing the results in their paper is sometimes misleading where the experiment settings are quite different.
>
> 4. I'm glad to see the NIAH results in your response. But still, it is not solid enough when only compared with CEPE.
>
>
> [1] Effective Long-Context Scaling of Foundation Models.
> [2] Data Engineering for Scaling Language Models to 128K Context.

---

### Official Review · Reviewer_wFyt · 2024-11-04

**Soundness:** 3
**Presentation:** 2
**Contribution:** 2
**Rating:** 6
**Confidence:** 3

**Summary:**

The paper presents SharedLLM, which uses two short-context LLMs (derived from the same model family like LLaMA-2) in a hierarchical structure to handle long contexts. My understanding is that this paper has made the following contribution:

- Integrating the concept of "Context Tree": A binary tree structure storing text at different granularities, with higher compression ratios at higher levels

- Architecture: A lower model compresses context into multi-grained representations, while an upper model performs language modeling using this compressed information

- Query-Aware Retrieval: For instruction-following tasks, uses similarity scoring to selectively expand relevant tree nodes

- Layer-wise Connection: Information transfer occurs only at lower layers to reduce computational overhead

Empirical results on language modeling tasks up to 128K tokens and various instruction-following benchmarks.

However, I am a bit concerned with the presentation and also the technical depth explanation.

**Strengths:**

- the basic empirical setup for evaluation in experimental studies is clean on standard benchmarks, from ppl to instruction-following tasks

- ablation studies demonstrate the effectiveness of the approach

**Weaknesses:**

1. In my personal opinon, this paper could benefit from the improving the presentation, specifically,

- In figure 1, it is hard for me to capture whether this is a encoder-decoder model or decoder-only model. The role of cross attention is not very clear. I would recommend authors to haven an overview of the model, and then dive into details, and use another figure to explain the context tree to avoid readers getting distracted. At least, you can mark which part is an encoder, and which part is a decoder.

- I really find it hard to understand what is "Tree Cross-attention" in Figure 1.

- The paper emphasizes on "self-injection" but this concept is nowhere in this central figure.

- there are notations of compression ratio but I am not so sure that they are well explained.

- Authors were arguing "information preservation" but I am not so sure what does this mean?could you elaborate on this and connect this with related work?

2. Experimental setup and results

- There are many hyperparameter setup of the trees constructed, but the intuitive why those hyperparameters were used/set are not well discussed.  See Q1

- What do you handle the drift or variable length during training and inference? See Q2

- Different values of M were set, what's the intuition and what's the best practice? See Q3

**Questions:**

Q1. How do you setup the values of hyperparameters of context tree? for example, their depth? are they sensitive to the inference tasks?

Q2. How do you take care of variable length during training and inference? The dynamic NTK and Yarn used the inference-time ratio on this, but I am not so sure about this in your method, thank you in advance as I am out of curiosity.

Q3. Robustness of different M values. M is an important model architecture base value, but I am not sure the robustness and meaning of setting different M values, and what is the best practice for this?

---

### Official Review · Reviewer_toYK · 2024-11-04

**Soundness:** 3
**Presentation:** 3
**Contribution:** 3
**Rating:** 6
**Confidence:** 5

**Summary:**

The paper introduces SharedLLM, a novel approach for extending the context window of large language models (LLMs) by using a hierarchical architecture that pairs two short-context LLMs. In SharedLLM, one model, called the lower model, acts as a compressor that processes past context into compact, multi-grained representations. The other, the upper model, serves as a decoder that integrates this compressed context with current text to predict future tokens in a context-aware manner. Information is passed from the compressor to the decoder through self-injection layers at specific levels, allowing efficient integration without extensive cross-attention.

**Strengths:**

1. Innovative Multi-Grained Context Extension: Introduces a unique approach to extend context windows using a compressor-decoder model architecture, which efficiently handles large context data.
2. Strong Experimental Results: Demonstrates superior performance on several long-context benchmarks, providing evidence of the model's robustness.
3. High Efficiency: Outperforms other methods in terms of speed and memory usage, making SharedLLM viable for large-scale applications.

**Weaknesses:**

1. The paper provides little information on how the model’s performance changes with different tree depths, compression ratios, and injection layers beyond the default settings. Since these parameters are key to achieving a balance between efficiency and accuracy, a sensitivity analysis would be beneficial.
2. While the paper introduces a query-aware retrieval policy in the context tree for efficient information extraction, it lacks a detailed analysis of how different retrieval policies affect SharedLLM’s performance. An ablation study comparing retrieval policies (e.g., different similarity metrics or selection thresholds) would enhance understanding and offer actionable tuning guidance for practitioners.

**Questions:**

1. What criteria guided the choice of retrieval policy in the context tree, and how sensitive is SharedLLM’s performance to different retrieval policy settings?
2. How does SharedLLM perform with different compression ratios, tree depths, and injection layer settings?
3. Have you considered testing SharedLLM with alternative context extension methods, such as position interpolation or memory-augmented architectures?

---

### Author Response · Authors · 2024-11-20
**General Response: A gentle reminder that we have updated the PDF file.**

Dear reviewers,

Thank you all for the efforts in reviewing this paper. This is a gentle reminder that we have slightly modified the PDF file and colored the modified text in red, which includes:
1) Change some colors in Figure 1 and add descriptions in the caption. This is to highlight the core dataflow of Self-injection in SharedLLM.
2) Add a few results on additional experiments in Appendix B, page 15. This is to further demonstrate SharedLLM's model and task generalizability.

If you find that we refer to specific lines or pages in our response, please download the latest version to ensure that the position of referred content is correct. We apologize for any incovenience this may cause.

We do not response to questions summarized from many reviewers since the questions raised by each reviewer focuses on different aspects. Please check the individual thread for targeted response. Thanks!

Best,

Authors

---

### Meta-Review · Area_Chair_zH8C · 2024-12-18

**Metareview:**

This paper proposes a method for longer context length using two LMs, which play role of a compressor and a decoder each other.

Longer context length without heavy computational costs and data for post-training are critical and important topic.
However, main concerns are too complicated method, experimental results not solid and insufficent in-depth analysis for supporting the efficacy of the proposed method, which are raised by most reviewers.

AC also agree to reviewers concerns, so recommends rejecting this paper.

**Additional Comments On Reviewer Discussion:**

Most reviewers pointed out its too complicated approach, insufficient and not solid experimental results and lack of in-depth analysis.
During the rebuttal, the authors tried to address these issues and failed to convince the reviewers.

Via AC-reviewer discussion, all reviewers and AC concluded that this  paper is not sufficient for ICLR quality.

---

### Decision · Program_Chairs · 2025-01-22

Reject